# Electrochemical Synthesis and Structural Characteristics of New Carbon-Based Materials Generated in Molten Salts

Deqiang Ji , Fajin Zhang, Zhiqiang Qiao, Jing Zhang, Hongjun Wu * and Guanzhong Wang

Key Laboratory of Energy Transformation and Utilization Technology, College of Chemistry and Chemical Engineering, Northeast Petroleum University, Daqing 163318, China
* Correspondence: hjwu@nepu.edu.cn

**Abstract:** This article is devoted to providing a new feasible route to realize carbon dioxide reduction and resource utilization. With the wide electrochemical window, high thermal stability and fast mass transfer rate of molten salt electrolyte, new carbon-based materials can be synthesized on the surface of the inexpensive Fe cathode. EDS (Electron-Dispersive-Spectroscope), SEM (Scanning-Electron-Microscope) and BET (Brunauer-Emmett-Teller) analyzers are selected to detect the critical element, microstructures and specific surface area of the new carbon-based materials generated via electrolysis. It is demonstrated that eutectic carbonates' electrochemical reduction, ranging from 450 °C to 750 °C prefers to produce carbons with no high-value structure. While carbon products are observed with honeycomb-like and platelet structures at 450 °C with an increase in current density. Additionally, the feedstock $CO_2$ could be converted into carbon-based materials with high value such as high surface area carbon, spherical carbon and cellular porous carbon production by optimizing the electrolysis parameters of temperature, current density and molten salt conformation. This paper shows a viable way for one-pot $CO_2$ utilization and facile production of micro-scale structure carbon materials, in line with the concept of sustainable development.

**Keywords:** molten salt electrochemistry; $CO_2$ utilization; carbon-based materials; microstructure characteristic; controllable synthesis

## 1. Introduction

As the world becomes aware of the link between the increasing $CO_2$ level in the atmosphere and the global warming crisis, the concept of $CO_2$ capture and conversion is progressively becoming one of the most frequently discussed topics. At present, researchers in various countries have made some achievements in the field of $CO_2$ capture and storage [1–3]. Besides being an important factor of the greenhouse effect, $CO_2$ is also widely recognized as the most abundant carbon-containing feedstock with the potential to be captured and supply chemicals production [4,5]. In the field of carbon materials synthesis, traditional metal catalysts represented by metal [6–8] or other preparation methods of carbon materials [9–11] have demonstrated excellent product forming performance. Arun Kumar summed up the techniques used for the synthesis of carbon materials with nanostructures by employing catalysts of natural feedstock [12]. The result is encouraging, but the consumption of natural materials (methane, ethylene, acetylene and petroleum gas) and the cost of catalysts are still big problems. There is still plenty of study challenges in the process of scaling up production due to the product control, catalyst life and conversion rate in the field of carbon materials generation.

Molten salt electrochemistry technology, an old but effective electrochemistry conversion method, is gradually attracting the attention of more scientists because of the wide electrochemistry windows and fast ion transfer rate. Subsequently, scientists have conducted more than half a century of research on synthesis of carbon materials via molten salt electrochemistry. It has been demonstrated that electrolysis temperature, electrode

potential and molten salt composition are factors for the morphology and nano surface area, and further affect properties and applications of carbon materials. The electrolyte of fluoride or chloride can be used as a reaction agent to produce solid carbon on the cathode [13,14]. However, the molar ratio of carbonate and the partial pressure of $CO_2$ in the atmosphere limit the technology's further contribution to greenhouse mitigation and $CO_2$ utilization. More importantly, there is an excellent capability that $CO_2$ can be directly reduced to final products consisting of the zero-valent element carbon (a transfer reaction of four electrons) in molten carbonates [15,16]. Significant advances in the field of $CO_2$ capture and chemical utilization have been made in recent years. Various nanostructures of new carbon-based materials have been synthesized successfully via molten carbonate electrolysis. As reported, Wang's group from Wuhan University electrolyzed the $Li_2CO_3$ + $Na_2CO_3$ + $K_2CO_3$ (43.5:31.5:25.0, mole ratio) ternary system at 500 °C and obtained amorphous carbon with an exciting result of a specific surface area—414 m$^2$/g. Using $Li_2CO_3$ + $K_2CO_3$ binary carbonate system (62:38, mole ratio) as the reaction medium, a variety of carbon nanostructures, such as sheet, ring, spheroid and amorphous, is obtained with a carbon deposition rate of 0.11 g/(cm$^2$·h).

In 2016, our group proposed and verified a new approach to generating gas fuel of $CH_4$ [17,18] and syngas [19,20] by adding an appropriate amount of hydroxides into molten carbonates, which further broadened the tunability and application prospect of the $CO_2$ and $H_2O$ co-electrolysis reaction. In the course of these studies, a fortuitous phenomenon was observed; that a small amount of black solid was deposited on the cathode surface, which turned out to be carbon-based materials. Due to the harsh reaction conditions and complex processes, the morphology control technology of carbon materials synthesized from $CO_2$ in molten salts has yet to make a significant breakthrough. It is an urgent scientific and technological problem in this field to effectively achieve the goal of high value-added carbon materials generation at nano and micro scales. In this paper, several lower cost carbonate components, involving $Na_2CO_3$, $K_2CO_3$ and divalent metal carbonates ($CaCO_3$ and $BaCO_3$), are employed to replace the expensive $Li_2CO_3$, to transform $CO_2$ into carbon materials with various morphologies with only one step. To the best of our knowledges, it is the first attempt to control the microstructure of synthetic carbon material in the molten salt system. This research is beneficial to mitigating global climate change and promoting the recycling of carbon resources.

## 2. Materials and Methods

Chemicals. Battery grade $Li_2CO_3$ (purity $\geq$ 99.99%) was supplied by China Shanghai Oujin Industrial Co., LTD. Other molten salt components, analytical-level $Na_2CO_3$, $K_2CO_3$, $CaCO_3$ and $BaCO_3$ (purity $\geq$ 99.0%), were provided by the Changchun Chemical Reagent Plant of China. Nickel wire (2 cm × 5 cm, active area = 10 cm$^2$) and galvanized iron wire (1 cm × 5 cm, active area = 5 cm$^2$) from Hebei SSMP company were employed as anode and cathode to provide sites for electrochemical redox reactions. A corrosion-resistant crucible, produced by Tangshan Kai Ping Porcelain Chemistry Factory with $Al_2O_3$ purity of 99%, was selected to contain molten salts at high temperatures.

Methods. $Li_2CO_3$, $Na_2CO_3$, $K_2CO_3$, $CaCO_3$ and $BaCO_3$ were dried under 150 °C in an air circulation oven for 12 h to avoid the effect of moisture, then mixed, ground evenly according to specific mass ratio and packed into the corundum crucible. The electrolyte was heated to 50 °C degrees above the melting point by the ceramic heating plate, equipped with a temperature controller, as Scheme 1 shows. After reaching the predetermined temperature, the molten salts electrolyzer needed to be stabilized for at least 0.5 h before electrolysis. The pretreated metal anode and cathode were inserted into the molten salts and current densities, or cell voltage was applied by high precision DC power supply (BK Precision 1715 A) across the two-electrode system. A lower current density of 15–25 mA/cm$^2$ was used to activate the Ni metal active site on the Fe cathode surface before expanding the current density to over 100 mA/cm$^2$ for rapid carbon deposition. Unless otherwise stated, the current densities described herein are expressed as cathode surface. The consumed

energy in heating and electrolysis could be substituted by wind, solar, tidal or other green renewable energy [21–23]. It was obvious that the cathode, removed from carbonate melts, was coated with a layer of solid black carbon after 1 A·h electric quantity. The deposited carbon was separated from the electrode surface by ultrasonic water bath initially. The carbon product was then washed in 1 M HCl for 48 h. The purpose of pickling was to remove the residual carbonates and dissolve the Fe or Ni that is reduced simultaneously with the carbon products. The pretreated carbon materials were to remove residual water content further via drying at 85 °C for at least 12 h.

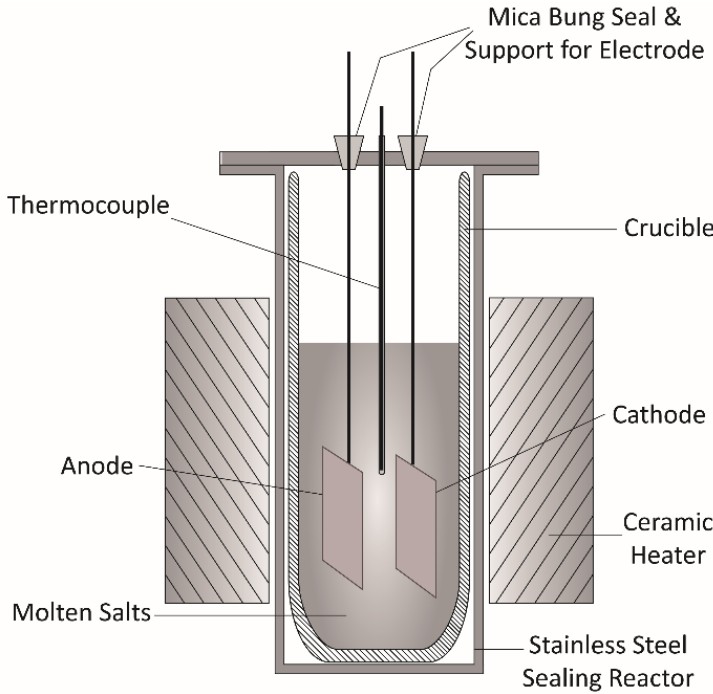

**Scheme 1.** Diagram of molten salt electrochemical reaction device.

Characterization. To accurately observe the microstructure and analyze the elemental composition of the obtained carbon materials, a Zeiss SIGMA scanning electron microscope (SEM) and a British Oxford Instruments energy dispersive spectrometer (EDS) were employed to characterize the carbon products. And the specific surface area of new carbon-based materials was revealed by Brunauer Emmett Teller Tristar-3020 from Micromeritics Instrument Corporation.

### 3. Results and Discussion

#### 3.1. $Li_2CO_3$ Splitting and $CO_2$ Transformation

In electrochemical study, liquid molten salt has become one of the most promising media in the electrochemical field because of its high conductivity, wide temperature range, low vapor pressure and excellent mass transfer and heat storage. Theoretically, all the carbonates can be used as electrolytes to produce carbon materials by $CO_2$ electrolysis. However, compared with other carbonates, monovalent cation carbonates ($Li_2CO_3$, $Na_2CO_3$ and $K_2CO_3$) have higher conductivity, better heat and mass transfer performance at the same temperature, which is conducive to promoting $CO_2$ reduction and reducing the reaction energy consumption.

The electrochemical reduction reaction of carbonate and hydroxide will occur at high temperature, and the thermal decomposition reaction will inevitably occur at the same time. As a competitive electrolytic reaction, the occurrence of thermal decomposition reactions will reduce the conversion of the target product and lead to energy independent loss. There has been some research reporting that the thermal stability of carbonates decreased with increasing temperature as Equation (1) describes [23,24]. As is shown in the picture, the

weight of $Li_2CO_3$ is almost constant under an open atmosphere (~0.03% $CO_2$), but dropped significantly over 800 °C. Thus, it indicates that the thermal decomposition reaction of $Li_2CO_3$ will not occur below 800 °C. Another obvious pattern is that prolonged reaction time leads to a more severe decrease in the weight of $Li_2CO_3$. It is evident that the weight decreases from 97.1% at the 6 h point to 85.9% at the 24 h point under 900 °C. According to the principle of chemical equilibrium, the reaction can be prevented from proceeding in the positive direction by increasing the concentration of the products in the system. It is indicated by the green line that the weight of $Li_2CO_3$ shows hardly any change, only 0.2% at 1000 °C after 6 h. Therefore, it is suggested that excess $CO_2$ rapidly combines with newly generated $Li_2O$ in molten salts and produces $Li_2CO_3$ again from the decreased weight loss. Similarly, we tried to increase the concentration of another product, $Li_2O$ to further slow the thermal decomposition of $Li_2CO_3$. Proof by the navy blue and the purple point in Figure 1, that $Li_2O$ can not only slow down the mass loss of $Li_2CO_3$ at high temperature, but also capture more $CO_2$ into the molten salt system (Equation (2)).

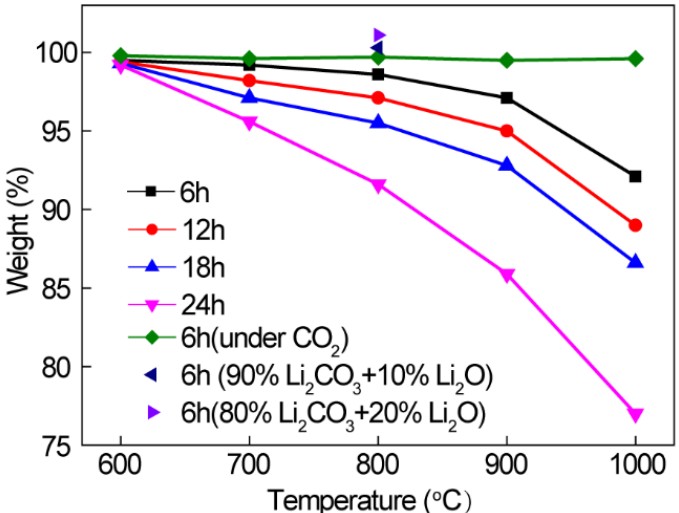

**Figure 1.** Thermodynamic stability of $Li_2CO_3$ (or $Li_2CO_3 + Li_2O$) at different temperatures.

$Li_2CO_3$ thermal decomposition:

$$Li_2CO_3 = Li_2O + CO_2 \qquad (1)$$

$CO_2$ dissolution:

$$Li_2O + CO_2 = Li_2CO_3 \qquad (2)$$

The experiment further verified the promotion effect of lithium oxide on the absorption of carbon dioxide in molten carbonates. No weight loss is observed in $Li_2CO_3 + Li_2O$ (90:10, weight ratio) at 800 °C for 6 h. More significantly, the purple point in Figure 1 shows an apparent mass gain in the system of 80% $Li_2CO_3$ + 20% $Li_2O$. As further presented, the metal oxide can act as an absorber of $CO_2$, but $Li_2O$ shows a more excellent property. It is suggested that molten carbonates, represented by $Li_2CO_3$, can transform $CO_2$ into carbon-based products and regenerate rapidly.

Molten $Li_2CO_3$ has over 10 moles of feedstock carbon per liter for electrochemical reaction at 750 °C. However, there is only 0.03–0.04% $CO_2$ in the air, about only $10^{-5}$ moles of reducible carbon per liter. It seems obvious that molten salt provides a million times more reducible tetrovalent carbon sites per unit volume than air. The multiplied reactant concentration contributes to promoting $CO_2$ conversion via the predetermined reaction path. The transformation process of $CO_2$ in carbonate melts includes the deposition of carbon on the cathode surface and the formation of oxygen on the anode surface. Because of the intermediate $Li_2O$, the molten electrolyte can be recycled and reused repeatedly. The above complete reaction process is described below.

Carbon deposition reaction:

$$CO_3^{2-} + 4e^- = C + 3O^{2-} \tag{3}$$

Oxygen formation reaction:

$$2O^{2-} = O_2 + 4e^- \tag{4}$$

$Li_2CO_3$ regeneration reaction:

$$O^{2-} + CO_2 = CO_3^{2-} \tag{5}$$

Overall reaction:

$$CO_2 = C + O_2 \tag{6}$$

As shown in Figure 2, a layer of solid carbon and melts is observed, following the chemical reaction, Equation (3), occurring on the surface of the galvanized iron cathode. There are some carbonates remaining on the surface of carbon products. It also explains why the acid pickling operation is essential for purification of carbon materials. Figure 2b shows that the purified carbon material after pickling—ultrasonic water cleaning—drying post-treatments. An EDS spectrogram is used to analyze the elemental composition and SEM is used to detect the microstructure of the obtained solid products. As displayed in Figure 2d, the molar ratio of the C element accounts for an overwhelming 90%, while only 10% of other elements remain. The obtained solid black powders are indeed carbon-based materials. The Ni element is derived from anodic corrosion during electrochemical reaction. The Cl element is derived from the hydrochloric acid pickling process, and the Au element comes from the gold spraying operation for EDS-SEM analysis. According to the previous reports, O is believed to originate from oxygen functional groups in carbon products [25,26].

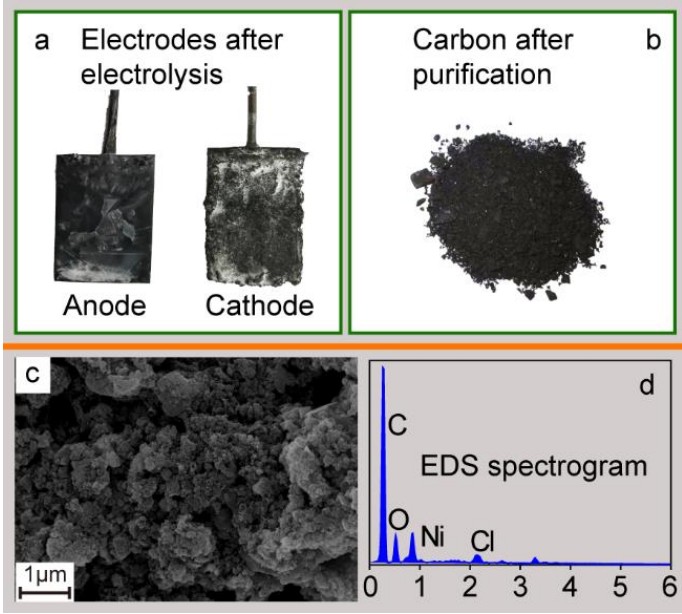

**Figure 2.** (**a**) Picture of Fe cathode surface after electrolysis; (**b**)carbon products after purification; (**c**) SEM of carbon at 750 °C under 200 mA/cm$^2$; (**d**) EDS result of the obtained carbon product.

### 3.2. Amorphous Carbon

In the synthesis process of new carbon-based materials, the carbon products obtained by electrolytic melting of $Li_2CO_3$ + $Na_2CO_3$ + $K_2CO_3$ under conventional conditions are mainly amorphous carbon. Figure 3a–f shows the micromorphology of carbon products obtained in the carbonate system of $Li_2CO_3$ + $Na_2CO_3$ + $K_2CO_3$ with an Fe cathode and

Ni anode at current density of 50, 100 or 200 mA/cm$^2$. As the picture below shows, the obtained carbon products are all amorphous carbon.

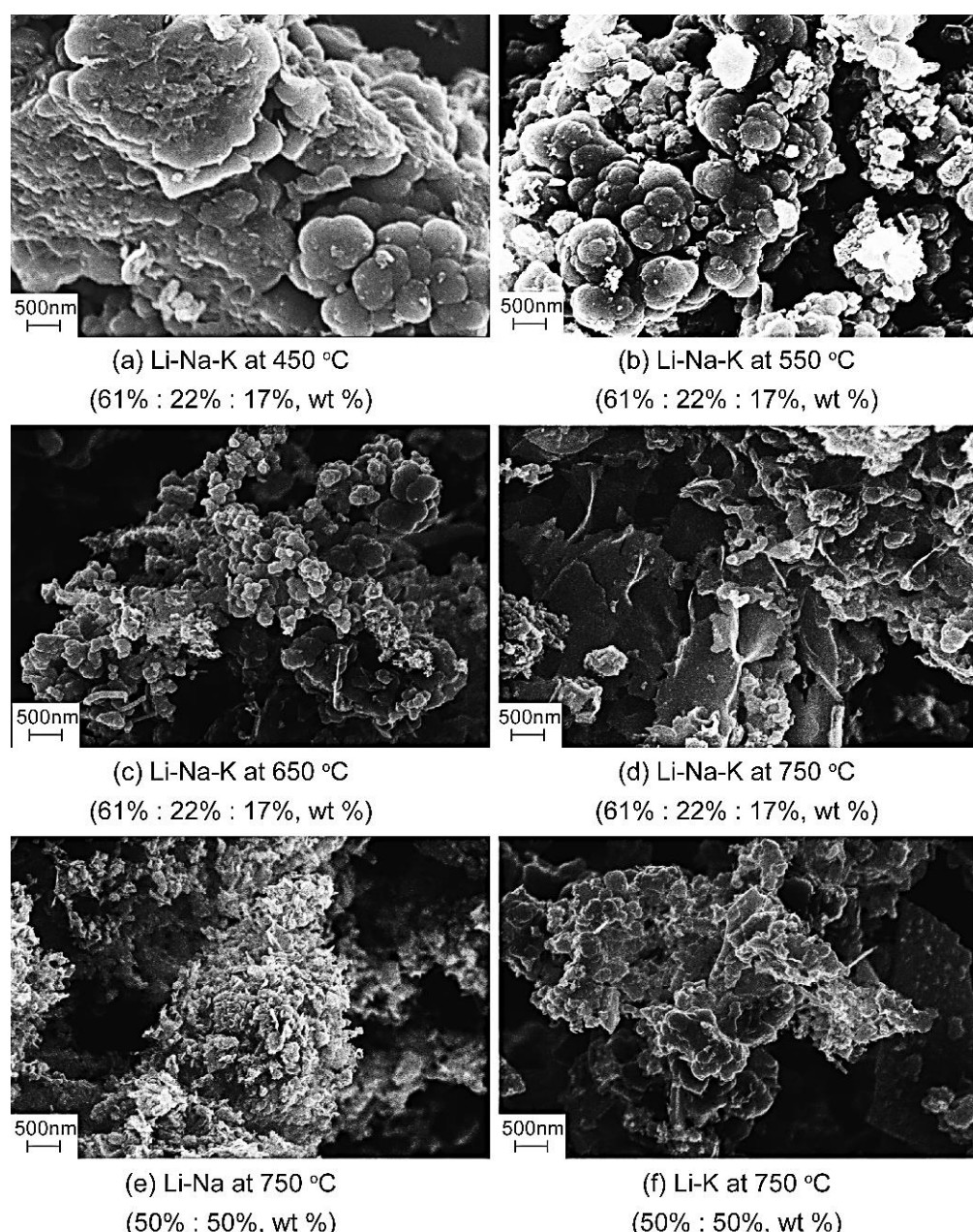

(a) Li-Na-K at 450 °C
(61% : 22% : 17%, wt %)

(b) Li-Na-K at 550 °C
(61% : 22% : 17%, wt %)

(c) Li-Na-K at 650 °C
(61% : 22% : 17%, wt %)

(d) Li-Na-K at 750 °C
(61% : 22% : 17%, wt %)

(e) Li-Na at 750 °C
(50% : 50%, wt %)

(f) Li-K at 750 °C
(50% : 50%, wt %)

**Figure 3.** Carbon products obtained from molten eutectic carbonates from 450 °C to 750 °C.

The operating temperature and current density/cell voltage have significant effects on the specific surface area of the prepared carbon materials. Lower operating temperature and higher current density/cell voltage are beneficial to increase the specific surface area of carbon-based materials. The results of carbon materials' specific surface area are displayed in Table 1. The specific surface area of obtained carbon is 127 m$^2$/g at 450 °C under 100 mA/cm$^2$ but drops sharply to 37.1 m$^2$/g at 550 °C under the same current density. The reason is that the increase in electrolysis temperature accelerates the carbon deposition reaction and promotes the formation of a dense carbon structure, which leads to a prominent decreasing specific surface area of carbon material. Besides, the top-down deposition rate of the cathode product slows down at low temperatures, resulting in a porous structure and a large specific surface area. For Li$_2$CO$_3$ + Na$_2$CO$_3$ + K$_2$CO$_3$ (61: 22:17, weight ratio) system electrolyzed at 750 °C, the specific surface area of formed carbon



is 28.2 m$^2$/g at 50 mA/cm$^2$ and increases to 51.7 m$^2$/g at 200 mA/cm$^2$. The reason for this structural change may be that the increase in current density leads to a higher voltage in the electrolysis cell. And the deposition of alkali metals on the carbon surface at higher cell voltage promotes the metal cation intercalation reaction, which is conducive to forming a loose and porous structure. The comparison of the products produced by a series of molten salt electrolytes with different compositions (NO.4, NO.7, NO.9) shows that the surface area of the carbon product is also affected by the composition and proportion of the electrolyte. The surface area of amorphous carbon generated from Li$_2$CO$_3$ + Na$_2$CO$_3$ + K$_2$CO$_3$ (33: 33: 33, weight ratio) is up to 412 m$^2$/g, but only 13.4 in Li$_2$CO$_3$ + K$_2$CO$_3$ (50%:50%). The reason is that under the same electrolysis temperature, the fluidity and conductivity of molten salt formulations are different, influencing the carbon deposition process. Different from the previously reported pure Li$_2$CO$_3$ electrolysis [27,28], carbon materials with a larger surface area are synthesized in a molten salt media containing less Li$_2$CO$_3$. More importantly, carbon materials with high surface area and large pore volume also have important applications in adsorption and material transport. From what has been discussed above, carbon materials with different specific surface areas can be synthesized by changing the electrolytic temperature, cell voltage/current density, electrolyte composition and ratio to meet diverse needs.

**Table 1.** The specific surface area of amorphous carbon with different electrolysis parameters.

| No. | Electrolyte (Weight Ratio) | Temperature (°C) | Current Density (mA/cm$^2$) | Specific Surface Area (m$^2$/g) |
|---|---|---|---|---|
| 1 | Li$_2$CO$_3$ + Na$_2$CO$_3$ + K$_2$CO$_3$ (61: 22:17) | 450 | 100 | 127 |
| 2 | Li$_2$CO$_3$ + Na$_2$CO$_3$ + K$_2$CO$_3$ (61: 22:17) | 550 | 100 | 89.5 |
| 3 | Li$_2$CO$_3$ + Na$_2$CO$_3$ + K$_2$CO$_3$ (61: 22:17) | 650 | 100 | 56.3 |
| 4 | Li$_2$CO$_3$ + Na$_2$CO$_3$ + K$_2$CO$_3$ (61: 22:17) | 750 | 100 | 37.1 |
| 5 | Li$_2$CO$_3$ + Na$_2$CO$_3$ + K$_2$CO$_3$ (61: 22:17) | 750 | 50 | 28.2 |
| 6 | Li$_2$CO$_3$ + Na$_2$CO$_3$ + K$_2$CO$_3$ (61: 22:17) | 750 | 200 | 51.7 |
| 7 | Li$_2$CO$_3$ + Na$_2$CO$_3$ + K$_2$CO$_3$ (33: 33:33) | 750 | 100 | 412 |
| 8 | Li$_2$CO$_3$ + Na$_2$CO$_3$ (50:50) | 750 | 100 | 37.6 |
| 9 | Li$_2$CO$_3$ + K$_2$CO$_3$ (50:50) | 750 | 100 | 13.4 |

### 3.3. Spherical Carbon

Because of the advantages of large specific surface areas, good fluidity and high mechanical strength, carbon spheres are widely used in gas storage [29], catalyst support [30] and drug delivery [31]. As previously described, the active nickel site originates from anodic corrosion, which can be used as an effective catalyst and for the growth of nucleated carbon products [28,32]. The formation of carbon material with special nanostructures in the molten carbonate system can be attributed to the nucleation of nickel particles [33]. Under the synergistic effect of high temperature and electric field, the surface of the nickel anode is oxidized to NiO. And the solubility of NiO in molten carbonate is very low, only $1 \times 10^{-5}$ mol/mol [34]. At the initial stage of electrolysis, a small current density is applied to the electrolytic system, about 15–25 mA/cm$^2$ for only 5 min. Subsequently, a current density of over 100 mA/cm$^2$ is applied to reduce the CO$_3^{2-}$ in the molten salt electrolyte to carbon. The metal nickel particles on the cathode surface act as nucleation sites in the process of carbon deposition. Carbon atoms accumulate on the surface of nickel particles, form bonds, and synthesize carbon materials with special structures.

Using the same electrode system as before, nickel wire (effective area 10 cm$^2$) and galvanized iron (effective area 5 cm$^2$) are employed as anode and cathode. The carbon products display a spherical structure generated in the molten salt containing calcium or barium carbonate via electrolysis at relatively high temperatures (650 °C or 750 °C). The microstructure of cathode carbon products, separated from the cathode surface in the electrolyte of Li$_2$CO$_3$ + CaCO$_3$ + Na$_2$CO$_3$, Li$_2$CO$_3$ + CaCO$_3$ + K$_2$CO$_3$, Li$_2$CO$_3$ + BaCO$_3$ + Na$_2$CO$_3$, Li$_2$CO$_3$ + BaCO$_3$ + K$_2$CO$_3$ and Li$_2$CO$_3$ + CaCO$_3$ + BaCO$_3$, is shown in Figure 4 in detail. Different from the amorphous carbon products synthesized in Li$_2$CO$_3$ + Na$_2$CO$_3$ + K$_2$CO$_3$ molten salt system, the morphology of carbon products of Li$_2$CO$_3$ + CaCO$_3$ + Na$_2$CO$_3$ and Li$_2$CO$_3$ + CaCO$_3$ + K$_2$CO$_3$ electrolytic systems at 750 °C shows a tufted sphere structure. The sphere diameter ranges from 300 to 500 nm, and the carbon spheres are superimposed.

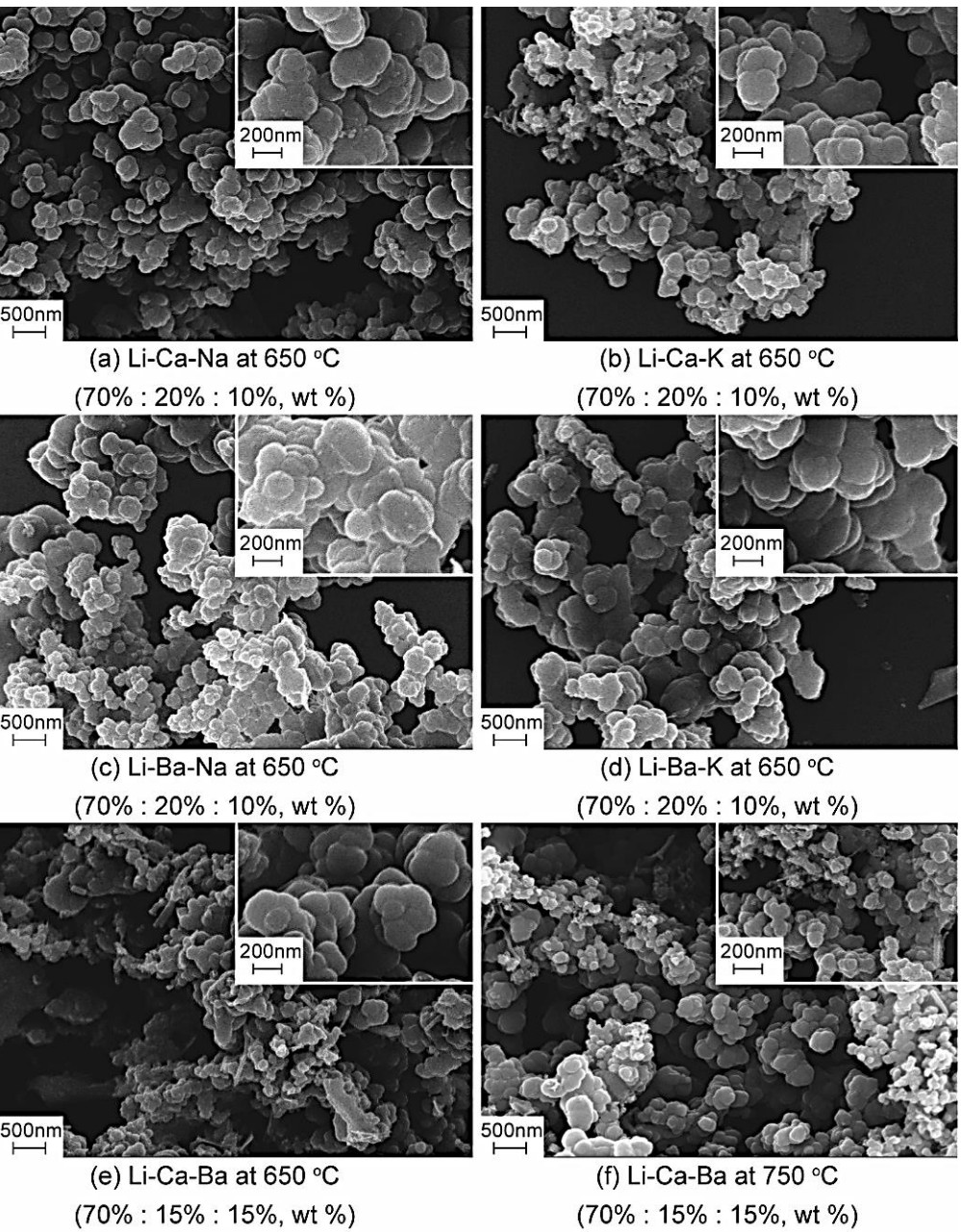

(a) Li-Ca-Na at 650 °C
(70% : 20% : 10%, wt %)

(b) Li-Ca-K at 650 °C
(70% : 20% : 10%, wt %)

(c) Li-Ba-Na at 650 °C
(70% : 20% : 10%, wt %)

(d) Li-Ba-K at 650 °C
(70% : 20% : 10%, wt %)

(e) Li-Ca-Ba at 650 °C
(70% : 15% : 15%, wt %)

(f) Li-Ca-Ba at 750 °C
(70% : 15% : 15%, wt %)

**Figure 4.** Carbon spheres obtained from different molten eutectic carbonates.

A current density of 25 mA/cm$^2$ is first applied to make nickel particles deposited on the cathode surface in the process of electrolysis. And then 200 mA/cm$^2$ is used to make the

carbon deposition reaction proceed at a fast speed. It has been proved in previous reports that $Na_2CO_3$ and $K_2CO_3$ are more likely to form corresponding alkali metals, which have a particular interference effect on the carbon deposition process catalyzed by nickel particles in the electrolysis process [17,32,33]. In molten salts containing divalent carbonates, the structure of carbon products is obviously changed from amorphous to spherical. Taking calcium carbonate as an example, CaO generated by the decomposition of $CaCO_3$ plays a certain interface modification role in the process of carbon deposition, leading to the electrolytic formation of carbon sphere morphology. During the deposition process, the interfacial modification effect of sufficient CaO not only leads to the formation of carbon spheroids, but also enhances the graphitization degree of carbon materials. The carbon nanostructures obtained in molten carbonate at high temperature or high current density exhibit high surface area and graphitization. Compared to the operating temperature of 650 °C, carbon spheres generated from $Li_2CO_3 + Ca_2CO_3 + Ba_2CO_3$ at 750 °C with gradient current density shows a more complete spherical shape, and the microscopic size is reduced to within 200 nm. In conclusion, the carbon spheres with regular morphology can be obtained by electrolysis of $Li_2CO_3 + Ca_2CO_3 + Ba_2CO_3$ in a high-temperature environment with the current density described above.

### 3.4. Cellular Porous Carbon

As mentioned above, lower electrolysis temperature and higher cell voltage/current density are beneficial for increasing the specific surface area of solid carbon-based products. Based on the above studies, $Li_2CO_3 + Na_2CO_3 + K_2CO_3$ electrolyte is selected as the primary electrolyte for the best process factors to synthesize carbon materials with high specific surface area. The results are shown in the following SEM pictures.

In $Li_2CO_3 + Na_2CO_3 + K_2CO_3$ system (mass fraction 61%:22%:17%), nickel-chromium alloy is used to replace nickel as the anode material and galvanized iron as the cathode. The effective area of the cathode and anode are both 5 $cm^2$. And the current density of 100, 200, 400 or 500 $mA/cm^2$ is applied for constant current electrolysis at different temperatures. Unlike the previously reported butterfly wing carbon sheet, the honeycomb structure shows a high degree of regularity, and its content is more than 80%, as displayed in Figure 5a–d. The aperture distribution of the honeycomb structure is broad, ranging from the nanometer to micron level. Honeycomb carbon, formed by stacking sheets of carbon together, has a compact structure and high specific surface area. According to BET, the specific surface area of cellular porous carbon, separated from the electrolysis product at 450 °C with 500 $mA/cm^2$ in $Li_2CO_3 + Na_2CO_3 + K_2CO_3$ electrolyte, is up to 198 $m^2/g$. The surface area of carbon material with honeycomb structure can be further improved by decreasing the electrolysis temperature or applying higher current density. In addition, honeycomb-like carbon materials with high specific surface area have essential applications in gas adsorption and material transportation. Under this electrolytic condition, the cell voltage is about 3.2 V, considerably higher than the theoretical deposition potential of $Li^+$, $Na^+$ and $K^+$. Consequently, at such high cell voltages, the carbon products deposition process is accompanied by an alkali metal deposition reaction. It is inferred that the co-deposition of alkali metals is a crucial factor in forming the honeycomb-like structure of carbon.

A Raman spectrometer is employed to detect the graphite crystal degree of cellular porous carbon, and the result is shown in Figure 6a. The Raman peak at 1340 $cm^{-1}$ (D band) corresponds to the defects and disorder-induced peaks in carbon materials. Another Raman peak at 1590 $cm^{-1}$ (G band) is produced by the stretching motion of $sp^2$ atom pairs in a carbon ring or long chain. The intensity ratio of the D band to the G band ($I_D/I_G$) is an important parameter to evaluate graphitization of carbon materials. For the cellular porous carbon obtained in the system of $Li_2CO_3 + Na_2CO_3 + K_2CO_3$, the $I_D/I_G$ value is about 0.8, indicating a high graphitization degree of the synthesized honeycomb carbon. An X-ray diffractometer is used to characterize the crystallinity to further reveal the structural features of the obtained carbon materials, and the result is displayed in Figure 6b. As the figure shows, there is a high intensity and sharp characteristic peak at 26°

(002), indicating that the graphite hexagonal crystal in honeycomb carbon is more regular and highly crystalline. Therefore, the carbon atoms in the obtained materials are mainly sp$^2$ hybridized. The electric capacity of carbon material with fewer defects is only about 360 mA·h/g [33]. It is inferred that the carbon material with higher sp$^2$ content is difficult to combine with Li$^+$ in a mechanism similar to embedding, resulting in lower capacitance.

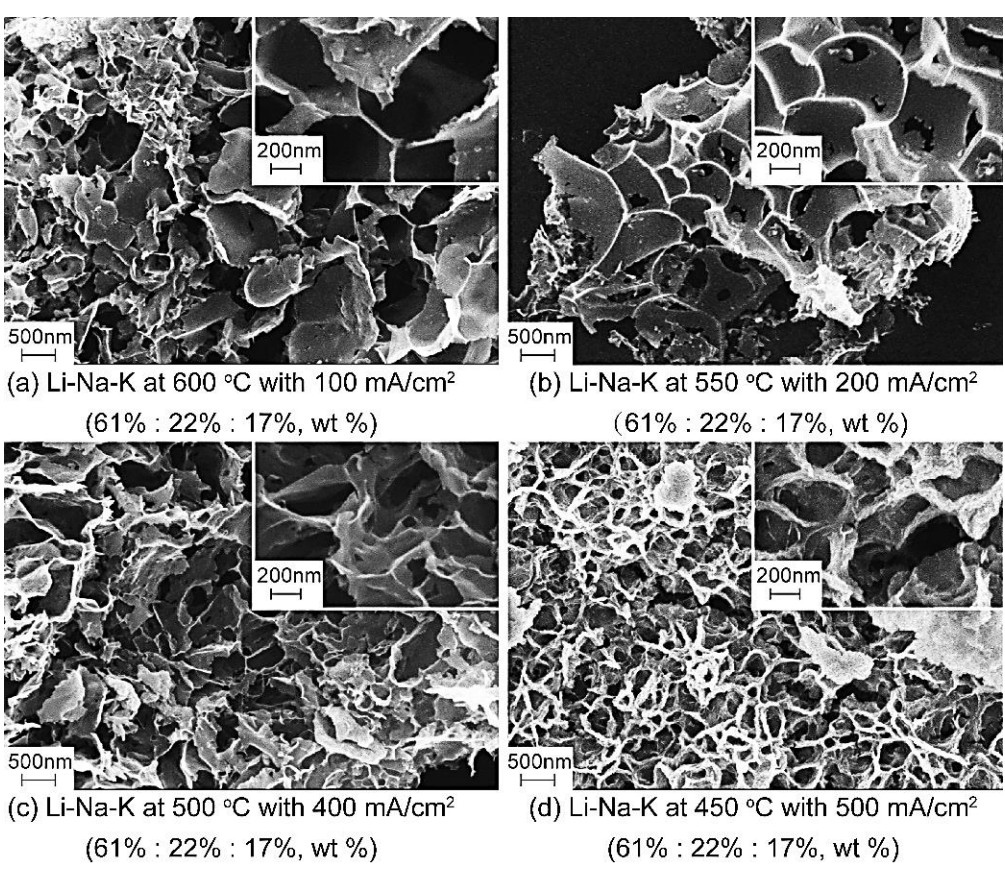

(a) Li-Na-K at 600 °C with 100 mA/cm$^2$
(61% : 22% : 17%, wt %)

(b) Li-Na-K at 550 °C with 200 mA/cm$^2$
(61% : 22% : 17%, wt %)

(c) Li-Na-K at 500 °C with 400 mA/cm$^2$
(61% : 22% : 17%, wt %)

(d) Li-Na-K at 450 °C with 500 mA/cm$^2$
(61% : 22% : 17%, wt %)

**Figure 5.** Cellular porous carbon obtained from Li$_2$CO$_3$ + Na$_2$CO$_3$ + K$_2$CO$_3$ eutectic carbonates under different conditions.

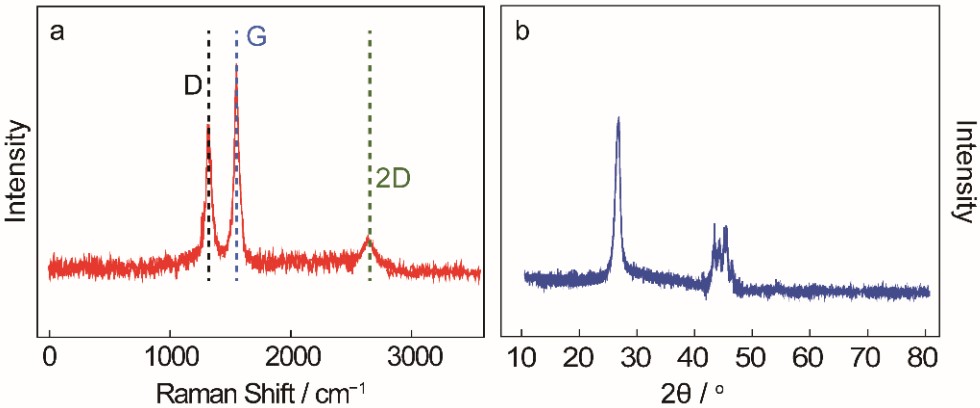

**Figure 6.** (**a**) Raman spectrum of cellular porous carbon. (**b**) XRD of cellular porous carbon.

As a new CO$_2$ utilization technology, molten salt electrochemical technology has attracted more attention in the field of carbon material synthesis. Different from arc excitation or vapor deposition, molten salt electrochemistry does not require noble metal catalysts and does not consume graphite or methane resources. In this paper, low-cost molten salt electrolyte with less Li$_2$CO$_3$ is employed to synthesize new carbon-based materials with

specific microstructure. More importantly, carbon materials with different micromorphology can be prepared by adjusting the electrolytic parameters without changing the electrode or reaction device. The purity of the carbon products is also relatively high and can meet application needs. In the following work, further improving the controllability degree and transformation efficiency of new carbon-based materials will be our research focus.

## 4. Conclusions

In this paper, a high-temperature molten carbonate electrolysis system is constructed for high value utilization of greenhouse gas $CO_2$. By adjusting the electrolyte composition, current density, electrolytic temperature and other parameters, $CO_2$ can be controllably transformed into carbon materials with various morphologies. Ternary $Li_2CO_3$ + $Na_2CO_3$ + $K_2CO_3$ molten carbonates prefer to form amorphous carbon in the temperature range of 450~750 °C with low current density. It is also proved that the surface area of new carbon-based products can be increased by decreasing operating temperature or improving current density/cell voltage. Replacing $Na_2CO_3$ or $K_2CO_3$ with divalent metal cation carbonate favors the generation of spherical carbon structures, probably due to the interfacial modification of sufficient CaO or interference effect of $Na^+$ and $K^+$. The $Li_2CO_3$ + $Na_2CO_3$ + $K_2CO_3$ electrolyte system provides an environment for the generation of high-purity honeycomb carbon materials at lower temperature with higher current density. In conclusion, carbon-based materials with varying structures can be synthesized by regulating the appropriate reaction parameters.

**Author Contributions:** Investigation, D.J., F.Z., Z.Q., J.Z. and G.W.; Project administration, H.W.; Writing—original draft, D.J.; Writing—review & editing, D.J. All authors have read and agreed to the published version of the manuscript.

**Funding:** This work was supported by the Natural Science Foundation of Heilongjiang Province (No. LH2021B005) and Guiding Innovation Foundation of Northeast Petroleum University (No. 2021YDL-09).

**Institutional Review Board Statement:** Not applicable.

**Informed Consent Statement:** Not applicable.

**Conflicts of Interest:** The authors declare no conflict of interest.

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
