# Peer review of "Electrochemical Synthesis and Structural Characteristics of New Carbon-Based Materials Generated in Molten Salts"

_applsci, doi:10.3390/app12199923_

Round 1
Reviewer 1 Report
The present work studies the electrochemical synthesis and structural characteristics of new carbon-based materials generated in molten salts. The authors used the following material characterization techniques: EDS (Electron-Dispersive-Spectroscope), SEM (Scanning-Electron-Microscope), and BET (Brunauer-Emmett-Teller). The work is relevant, but it does not present any technological innovation. The result is well written, but there are improvements to be made in the writing of the English language, in addition to correcting some typos and errors. Below I leave my suggestions for improving the work:
i) It is necessary to improve the Introduction. It is vital to write a paragraph describing the novelty and technological innovation of the present work;
ii) In addition to the EDS, it is crucial to add another technique for chemical characterization of the structures, such as FTIR or XPS;
iii) In Results and Discussions, the authors present the results but without a discussion with other works in the literature. I suggest that the authors provide this improvement urgently;
iv) This Journal is about applications, but the authors do not show the possible applications of the different structures of the materials developed in the present work;
v) The literature used is very poor, so the manuscript needs a more detailed literature review to support the work;
Therefore, in the form presented, the manuscript cannot be accepted for publication.
Reviewer 2 Report
The authors have chosen a worthy topic. They have reported the possibility of CO2 utilisation and production of valuable carbon materials. The paper can be accepted but with some minor inclusions.
1. Please discuss the importance of BaCO3 and CaCO3 in the introduction and mention them in the experimental section. They suddenly appear in the results section.
2. It is better to include the experimental setup in the paper.
3. Please include the anode's picture in figure two as well.
4. As you have used SEM, were you able to estimate the thickness of Ni deposits on the cathode and did it play any role in the efficiency of the process?
5. There are some minor grammatical and spelling errors, please kindly correct them.
Irrespective of the above concerns, I believe that the manuscript is fit to be published in Applied Sciences.
Reviewer 3 Report
The manuscript, ‘Electrochemical synthesis and structural characteristics of new carbon-based materials generated in molten salt’ by Ji et. al. provides a easy route to utilize the reduced carbon dioxide for the growth of carbon products. Reviewer has found few issues with the manuscript, as well as few comments to improve the scientific standard of the manuscript. The manuscript could be accepted after major revision, by addressing the following comments:
1. Authors have placed the same optical image in Figure 2b, as of Figure 2c of the following previously published paper, Appl. Sci. 2022, 12(17), 8874; https://doi.org/10.3390/app12178874. As it should be different and authors should follow the scientific ethics and journals regulations, before submitting anything for publications.
2. Introduction is very generic, and could be improved in comparison to the previously published work by the authors, Appl. Sci. 2022, 12(17), 8874; https://doi.org/10.3390/app12178874. Authors could provide more details on the points mentioned in line 34-36, with appropriate references. And other results related to utilization of natural resources for carbon materials, for example there has been reports related to natural catalyst utilizations; Physchem 2021, 1(1), 4-25; https://doi.org/10.3390/physchem1010002, Carbon 2022, 196, 510-524, https://doi.org/10.1016/j.carbon.2022.05.025.
3. What is the origin of Ni in the carbon products, as shown in Figure 2d, EDS results.
4. Line 188-190, As previously described ‘the active nickel site originates from anodic corrosion, which can be used as an effective catalyst and for the growth of nucleated carbon products’. There was no previous description on this, so please specify and provide a suitable reference. Next, yes it is true that Ni can act as effective catalyst, but in the manuscript it not specifically mentioned the role of Ni for the growth of Carbon materials, please specify the role of Ni.
5. Line 247, provide the reference.
6. It has been concluded that, the surface area of new carbon-based products can be increased by decreasing operating temperature or improving current density/cell voltage. Than how does it affect the yield of the carbon materials.
7. How much was the yield of the obtained products ?
8. Authors are required to provide at least one structural or vibrational spectroscopy results to confirm the degree of graphitization of the obtained carbon products.
9. Authors should explain the possible applications of the obtained carbon materials.
Reviewer 4 Report
Dear authors
I have overall enjoyed article reading. The topic discussed by the authors is interesting to the audience and highly relevant to current research trends. Considering the quality of the manuscript, I have recommended major changes which are next listed:
Major changes:
The usage of molten salts have been thoroughy used in the past to generate carbon-based materials, could you please expand the literature review? Similarly, could you include a discussion about how relevant your findings are when compared with previous literature results? This new section can be introduced before the conclussions.
Minor changes
Line 31: Although it may be obvious for an average reader, please provide a definition for the acronym C1.
Figures 3 and 5: If possible, it would be great to improve the resolution of the SEM images; this recommendation specially holds for Figures 5(a) and Figure 5(c).
Author Response
I have overall enjoyed article reading. The topic discussed by the authors is interesting to the audience and highly relevant to current research trends. Considering the quality of the manuscript, I have recommended major changes which are next listed:
*On behalf of my co-authors, we are very grateful to you for giving us an opportunity to revise our manuscript. we appreciate you very much for your positive and constructive comments and suggestions on our manuscript entitled “Electrochemical synthesis and structural characteristics of new carbon-based materials generated in molten salts”.
Major changes:
The usage of molten salts have been thoroughy used in the past to generate carbon-based materials, could you please expand the literature review? Similarly, could you include a discussion about how relevant your findings are when compared with previous literature results? This new section can be introduced before the conclussions.
*As per the reviewer’s suggestion, we have added a description of the innovation and technical advantages of this paper from Line 326-334.
Minor changes
Line 31: Although it may be obvious for an average reader, please provide a definition for the acronym C1.
*We thank the reviewer for pointing this out. The “C1 feedstock” has been modified to “carbon-containing feedstock”.
Figures 3 and 5: If possible, it would be great to improve the resolution of the SEM images; this recommendation specially holds for Figures 5(a) and Figure 5(c).
*As per the reviewer’s suggestion, Figures 3 and 5 have been reset to improve resolution.
Round 2
Reviewer 1 Report
The authors made the suggested changes, so the work is ready to be published.
Author Response
We are very grateful for your comments regarding our manuscript “Electrochemical synthesis and structural characteristics of new carbon-based materials generated in molten salts”.
Reviewer 3 Report
Authors have significantly improved the content and description in the revised manuscript. The reviewer is satisfied with the comments response by the authors. The maniscript is acceptable for publication. However, authors need to read the manuscript carefully to correct the minor issues before submitting the proof read, such as Figure 6, captions are not coorectly written.
Author Response
*We are very sorry for such a low-level error. And all the minor issues, such as the captions of Fig.6, have been corrected carefully.
Reviewer 4 Report
Considering your reply below (between brackets):
"*As per the reviewer’s suggestion, we have added a description of the innovation and technical advantages of this paper from Line 326-334."
I could not find the introduced text in lines 326-334. Please check line numbering as generated by the mpdi platform
Author Response
*We are very sorry for this mistake. we have added a description from Line 378-388. And the additions are as follows "
As a new CO2 utilization technology, molten salt electrochemical technology has attracted more attention in the field of carbon material synthesis. Different from arc excitation or vapor deposition, molten salt electrochemistry does not require noble metal catalysts and not consume graphite or methane resources. In this paper, low-cost molten salt electrolyte with less Li2CO3 is employed to synthesize new carbon-based materials with specific microstructure. More importantly, carbon materials with different micromorphology can be prepared by adjusting the electrolytic parameters without changing the electrode or reaction device. The purity of the carbon products is also relatively high and can meet the application needs. In the following work, further improving the controllability degree and transformation efficiency of new carbon-based materials will be our research focus.
"